# DiRA: Nuclear Norm Dynamic Rank Adaptation for Large Language Models

## Abstract

Parameter-Efficient Fine-Tuning (PEFT) methods, particularly Low-Rank Adaptation (LoRA), have become a standard paradigm for adapting Large Language Models (LLMs) to specific tasks. However, standard LoRA implementations use a fixed, uniform adaptation rank across all layers, a static allocation that fails to capture the varying contributions of different layers. In this work, we introduce DiRA, which learns layer-adaptive ranks by penalizing the nuclear norm of the weight update matrix $\Delta W$ for each layer. While extensive experiments show that DiRA matches or surpasses fixed-rank LoRA baselines across tasks, its primary contribution is methodological and scientific. Using DiRA as a probe, we uncover a mechanism of catastrophic forgetting in continual learning: forgetting is frequently accompanied by pronounced changes in the rank landscape. Building on this insight, we propose a new strategy that treats the previously learned rank landscape as a prior and, with only a small amount of data, regularizes current updates to retain newly acquired knowledge while recovering old-task memory, thereby mitigating forgetting. Taken together, these results position DiRA both as an efficient PEFT method and as a principled approach for understanding—and mitigating—forgetting in LLMs.

## 1 Introduction

Recent advancements in pre-trained large language models (LLMs), such as Llama (Touvron et al., 2023), Qwen (Bai et al., 2023), and GPT-5 (OpenAI, 2025), have significantly enhanced performance across a wide range of natural language processing tasks. Traditionally, adapting these powerful models for specific applications has required comprehensive fine-tuning, a process that updates all model parameters. However, given the colossal scale of modern LLMs, full fine-tuning is often computationally prohibitive, particularly in resource-constrained environments.

To address this challenge, Parameter-Efficient Fine-Tuning (PEFT) methods have emerged as a practical alternative. These techniques adapt LLMs by updating only a small fraction of their parameters while keeping the majority of the original architecture frozen (Lester et al., 2021; Liu et al., 2021; Hu et al., 2022; Liu et al., 2024). Among these, Low-Rank Adaptation (LoRA) (Hu et al., 2022) has become a prominent approach. LoRA injects trainable low-rank matrices into the model, parameterizing the weight update $\Delta W$ as a product of two smaller matrices. By only training the two smaller matrices, LoRA dramatically reduces the number of trainable parameters and associated computational costs compared to updating the full-rank weight matrix $W$.

Despite its efficiency, LoRA and many of its variants (Lialin et al., 2023; Hayou et al., 2024) are constrained by a critical limitation: they pre-specify a uniform, fixed rank $r$ for every adapted weight matrix. This design overlooks the heterogeneous importance of different matrices across a model's layers and modules. As prior work has shown (Zhang et al., 2023; Shinwari & Usama, 2025), allocating a larger parameter budget to more critical weight matrices can substantially improve model performance. Conversely, assigning excess parameters to less important matrices yields diminishing returns. Given a fixed parameter budget for fine-tuning, an optimal strategy should allocate resources discriminately. The uniform allocation employed by LoRA, as well as other methods like adapters and prefix-tuning (Li & Liang, 2021), is therefore inherently suboptimal.

To address fixed-rank limitations, two complementary lines of work have emerged. First, direct dynamic-rank methods adjust rank during training. AdaLoRA (Zhang et al., 2023) replaces fixed-

rank LoRA with an SVD-inspired parameterization $\Delta W = P\Lambda Q^{\top}$, where the diagonal of $\Lambda$ encodes learnable importance scores; low-importance components are iteratively pruned, yielding data-driven ranks but adding training-time pruning and orthogonality regularization complexity. ARD-LoRA (Shinwari & Usama, 2025) automates per-layer/per-head rank selection via differentiable scaling/gating, typically combining sparsity-inducing priors and smoothness regularization to stabilize rank transitions, enabling continuous, fine-grained adaptation at the cost of extra controls and more elaborate objectives or meta-optimization. Second, sparsity-based methods reduce effective (numerical) rank indirectly: LoSA (Huang et al., 2025) and SMT (He et al., 2025) impose sparsity on $\Delta W$ or on coefficients of low-rank factors, zeroing non-critical directions and concentrating capacity where it matters while implicitly lowering the numerical rank. These schemes often simplify parameterization and deployment (via pruning and sparse kernels) but must address challenges in learning sparse masks, tuning the sparsity–performance trade-off, and coordinating sparsity budgets across layers.

In this work, we propose Dynamic Rank Adaptation (DiRA), a novel PEFT framework that reformulates rank allocation as a direct, regularized optimization problem. Our key insight is to decompose the LoRA update as a sum of rank-1 components and apply nuclear norm regularization (Fazel, 2002; Candes & Recht, 2012; Recht et al., 2010; Cai et al., 2010). This mechanism encourages the model to drive entire rank-1 components to zero during standard backpropagation, allowing the optimal, layer adaptive rank to emerge organically as an intrinsic property of the learned solution. Unlike complex external processes in prior work, DiRA performs automatic model selection within the fine-tuning process itself, effectively tailoring its complexity to the available data without additional hyperparameter tuning or iterative procedures.

Leveraging DiRA, we further uncover an empirical phenomenon in continual learning: catastrophic forgetting (McCloskey & Cohen, 1989; French, 1999; Kirkpatrick et al., 2017; Zenke et al., 2017; Grossberg, 1987) is frequently accompanied by pronounced changes in the rank landscape. Motivated by this observation, we introduce a new strategy that uses the rank landscape learned on prior tasks as a prior: with a small subset of data from old tasks, we regularize the current update to preserve the previous rank landscape while acquiring new knowledge, thereby balancing performance on old and new tasks (Rebuffi et al., 2017; Chaudhry et al., 2019; Buzzega et al., 2020; Rolnick et al., 2019). We note that layer/head-wise heterogeneity in importance (Michel et al., 2019) further justifies preserving a structured rank landscape when transferring across tasks.

Our contributions are:

1. We introduce DiRA, a new end-to-end PEFT framework that performs dynamic, layer-wise rank selection via nuclear norm regularization.

2. We uncover a mechanism-level connection in continual learning: forgetting is often accompanied by pronounced changes in the model's rank landscape.

3. We propose a rank landscape prior, applied using a small subset of data from previous tasks, that stabilizes old task performance while preserving performance on new tasks.

## 2 RELATED WORKS

### 2.1 PARAMETER-EFFICIENT FINE-TUNING

Adapting large language models (LLMs) to downstream tasks through full fine-tuning is often computationally prohibitive due to the vast number of parameters involved (Han et al., 2024; Xu et al., 2023). To address this challenge, Parameter-Efficient Fine-Tuning (PEFT) methods have been proposed to reduce the number of trainable parameters while preserving model performance. Prominent PEFT techniques include Adapter Tuning (Houlsby et al., 2019), which inserts small, trainable modules between the layers of a pre-trained model; Prefix-Tuning (Li & Liang, 2021), which prepends a sequence of trainable virtual tokens to the input; and Prompt-Tuning (Lester et al., 2021), which learns a soft prompt to condition the model's behavior.

## 2.2 Low-Rank Adaptation (LoRA) and Its Variants

A particularly influential PEFT method is Low-Rank Adaptation (LoRA) (Hu et al., 2022). LoRA freezes the pre-trained weights and injects trainable low-rank matrices into the Transformer layers, thereby drastically reducing the number of parameters required for adaptation. The core idea is to represent the weight update $\Delta W^l$ as a low-rank product $\Delta W^l = B^l A^l$, where $l$ denotes which layer of the model. As an extension, DoRA (Liu et al., 2024) decomposes weight updates into magnitude and direction, updating the directional component in the same manner as LoRA.

The standard LoRA implementation utilizes a fixed, uniform rank across all adapted layers. However, this may be suboptimal, as different layers might require different capacities for task adaptation. This has led to the development of methods for dynamic rank allocation. For instance, AdaLoRA (Zhang et al., 2023) dynamically allocates the parameter budget to weight matrices based on their importance scores. More recent works like ARD-LoRA (Shinwari & Usama, 2025) and RankAdaptor (Zhou et al., 2024) also explore automated, layer-wise rank determination. Our proposed method, which we term Dynamic Rank Adaptation (DiRA), contributes to this emerging direction. It formulates rank selection as a regularization problem, using a nuclear norm penalty to the LoRA update, thereby allowing each layer to learn its own optimal rank from data.

## 2.3 Nuclear Norm and Low-Rank Regularization

The nuclear norm (also called the trace norm or the Schatten-1 norm, denoted $\| \cdot \|_*$) is a widely used convex surrogate for matrix rank in low-rank recovery, matrix completion, and many machine-learning applications. For a matrix $X \in \mathbb{R}^{m \times n}$ with singular values $\sigma_1(X) \geq \cdots \geq \sigma_{\min(m,n)}(X)$, the nuclear norm is defined as

$$\|X\|_* := \sum_{j=1}^{\min(m,n)} \sigma_j(X), \tag{1}$$

and serves as a natural convex relaxation of the nonconvex rank function; classical results show that minimizing the nuclear norm under appropriate observation models can recover ground-truth low-rank matrices (Fazel, 2002; Recht et al., 2010; Cai et al., 2010; Candes & Recht, 2012).

The nuclear norm admits several useful variational characterizations that provide both intuition and practical algorithmic routes. A common factorization-based identity is

$$\|X\|_* = \min_{X=UV^\top} \frac{1}{2}\Big(\|U\|_F^2 + \|V\|_F^2\Big), \tag{2}$$

where $U \in \mathbb{R}^{m \times r}$ and $V \in \mathbb{R}^{n \times r}$, and $r$ can be taken sufficiently large (e.g., $r \geq \operatorname{rank}(X)$) (Srebro et al., 2004; Mazumder et al., 2010). Another equivalent representation (Chandrasekaran et al., 2012), which is directly relevant to componentwise (rank-1) parameterizations, is the decomposition form

$$\|X\|_* = \min_{X=\sum_i u_i v_i^\top} \sum_i \|u_i\|_2 \|v_i\|_2, \tag{3}$$

which interprets the nuclear norm as the minimal sum of products of Euclidean norms over all rank-1 decompositions of $X$. This last form explains why penalizing per-component products $\|u_i\|_2\|v_i\|_2$ encourages low-rank solutions.

In this work, our regularizer is directly inspired by the variational decomposition above: we penalize each rank-1 component by the product $\|u_i\|_2\|v_i\|_2$. This design preserves the low-rank inducing bias of the nuclear norm while avoiding repeated full SVDs during training, yielding a practical compromise between theoretical motivation and computational tractability. We empirically compare our factorized penalty against other approximations and baselines in Appendix B.2.

## 3 Method

### 3.1 Dynamic Rank Adaptation (DiRA)

For each layer $l$, standard LoRA modifies the forward pass of a pre-trained weight matrix $W_0^l$ as $h^l = W_0^l x + B^l A^l x$, where the rank $r$ of matrices $B^l$ and $A^l$ is a fixed hyperparameter. To overcome this static rank limitation, we propose a dynamic approach DiRA.

**Rank-1 Decomposition.** We first re-conceptualize the LoRA update matrix, $\Delta W^l = B^l A^l$, as a sum of up to $r$ rank-1 matrices:

$$\Delta W^l = \sum_{i=1}^{r} B_{:,i}^l A_{i,:}^l. \tag{4}$$

Here, each $B_{:,i}^l$ is a column vector in $\mathbb{R}^{d_2 \times 1}$ and each $A_{i,:}^l$ is a row vector in $\mathbb{R}^{1 \times d_1}$. The hyperparameter $r$ defines the maximum possible rank for any layer, with the effective rank being learned from the data.

**Dynamic Rank Allocation.** To encourage the model to learn a dynamic rank structure, we add a nuclear norm regularization term that penalizes the Frobenius norm of each per-component product $(B_{:,i}^l, A_{i,:}^l)$. Formally, we define the total loss as

$$
\begin{aligned}
\mathcal{L}_{\text{total}} &= \mathcal{L}_{\text{task}} + \lambda \sum_{l=1}^{k} \|\Delta W^l\|_* \\
&= \mathcal{L}_{\text{task}} + \lambda \sum_{l=1}^{k} \sum_{i=1}^{r} \|B_{:,i}^l\|_2 \|A_{i,:}^l\|_2,
\end{aligned}
\tag{5}
$$

where $\mathcal{L}_{\text{task}}$ is the primary task loss (e.g., cross-entropy), $\|\cdot\|_*$ and $\|\cdot\|_2$ denote the nuclear and Euclidean norms, respectively, and $\lambda$ is a regularization coefficient. Directly regularizing the LoRA update matrices $\{\Delta W^l\}_{l=1}^{k}$ by the nuclear norm $\|\Delta W^l\|_*$ provides principled, spectrum-level capacity control, but computing $\|\Delta W^l\|_*$ typically requires SVD of a $d_2 \times d_1$ matrix at each layer and frequently during training, which is computationally expensive. To avoid repeatedly forming large SVDs, we exploit a simple upper bound that leverages the rank-1 structure of the LoRA decomposition. See the Appendix E for more details.

## 4 EXPERIMENTS

To evaluate our method across a diverse set of capabilities, we conduct experiments on 2 distinct task categories: commonsense reasoning and open-domain dialogue.

**Commonsense Reasoning.** We use a comprehensive benchmark aggregated by (Hu et al., 2023), which combines eight different datasets. The consolidated training set contains 170,300 query–answer pairs, and we use 120 examples for validation. The benchmark spans a broad range of reasoning skills, including Yes/No QA (BoolQ (Clark et al., 2019)); physical and social commonsense (PIQA (Bisk et al., 2020), SIQA (Sap et al., 2019)); natural language inference and completion (HellaSwag (Zellers et al., 2019), WinoGrande (Sakaguchi et al., 2021)); and science and multi-step reasoning (ARC-c, ARC-e (Clark et al., 2018), OBQA (Mihaylov et al., 2018)). Detailed statistics for each dataset are provided in Appendix C.

**Open-domain Dialogue Generation.** We employ the ConvAI2 dataset (Dinan et al., 2019), which consists of 17,878 multi-turn training conversations and 1,000 for testing. In this task, models must generate conversational responses conditioned on a speaker's persona, which is described in 4–5 sentences. Following the experimental setup of prior work (Liu et al., 2020; Song et al., 2021; Huang et al., 2023; 2024b), we use a self-persona setting where the model is only given access to the speaker's own persona profile.

### 4.1 EXPERIMENTAL SETTINGS

**Evaluation metrics.** We use accuracy as the primary metric for the commonsense-reasoning datasets, computed separately for each subtask. For each test instance the model decodes an answer from the prompt; we then scan the model's output for task-specific answer keywords (e.g., true / false for BoolQ). The first occurrence of a matching keyword is taken as the model's response; if no relevant keyword appears, the response is scored as incorrect. This deterministic keyword-matching rule is applied uniformly across all eight subtasks and follows prior work (Hu et al., 2023; Liu et al.,

2024). For ConvAI2 we report BLEU (Papineni et al., 2002) and BERTScore (Zhang et al., 2019). For the mathematical reasoning task we again use accuracy.

**Baselines.** We compare DiRA against representative prompt-based and low-rank adaptation methods. Prompt-based baselines include P-Tuning (Liu et al., 2021). Low-rank adaptation baselines include LoRA (Hu et al., 2022), DoRA (Liu et al., 2024), and LoRA+ (Hayou et al., 2024). Experiments are conducted with the open-source models Llama2-7B (Touvron et al., 2023) and Llama3-8B (Dubey et al., 2024).

**Implementation Details.** We follow the training setup of (Liu et al., 2024) with only minor adjustments to the learning rate. DiRA is implemented on Llama2-7B and Llama3-8B using rank $r = 32$. Optimization is performed with AdamW (Loshchilov & Hutter, 2017) at a base learning rate of $1 \times 10^{-4}$, with a linear warm-up over the first 100 steps. For the commonsense-reasoning benchmarks we fine-tune each model for 3 epochs and evaluate on the validation set every 80 steps to select the best checkpoint.

We apply LoRA, DoRA, LoRA+ and DiRA to the query, key, and value projection matrices as well as the attention down- and up-projection linear layers. To ensure fair comparison, all methods are configured to have the same or comparable numbers of trainable parameters; prompt-based baseline P-Tuning is adjusted so their trainable-parameter budgets are comparable as well. Detailed hyperparameter settings and parameter budgets are provided in Appendix C. The only deviations from the above schedule is: ConvAI2 experiments use 1 epoch.

## 4.2 RESULTS ON COMMONSENSE REASONING TASKS

Table 1: Accuracy comparison among various PEFT methods on commonsense reasoning datasets. Results for ChatGPT, P-Tuning and DoRA are sourced from (Huang et al., 2024a). **Bold**: the best.

| Model | Method | BoolQ | PIQA | SIQA | ARC-c | ARC-e | OBQA | HellaS | WinoG | Average |
|-------|--------|-------|------|------|-------|-------|------|--------|-------|---------|
| ChatGPT | - | 73.10 | 85.40 | 68.50 | 79.90 | 89.80 | 74.80 | 78.50 | 66.10 | 77.01 |
| Llama2-7B | P-Tuning | 58.75 | 36.02 | 0.20 | 0.17 | 1.98 | 0.80 | 0.01 | 0.00 | 12.24 |
| | LoRA | 70.21 | 83.26 | 79.60 | 74.00 | 87.09 | 84.40 | 88.10 | 84.79 | 81.43 |
| | LoRA+ | 71.03 | 84.36 | 79.55 | 75.25 | 87.43 | **86.20** | 88.72 | 84.18 | 82.09 |
| | DoRA | 71.80 | 83.70 | 76.00 | 68.20 | 83.70 | 82.40 | **89.10** | 82.60 | 79.69 |
| | AdaLoRA | 71.19 | 85.36 | 79.96 | 75.19 | **88.76** | **86.20** | 88.95 | 85.11 | 82.60 |
| | DiRA | **72.84** | **85.47** | **80.45** | **75.68** | 88.13 | 85.80 | 87.97 | **85.64** | **82.75** |
| Llama3-8B | P-Tuning | 59.97 | 11.64 | 8.19 | 7.42 | 8.63 | 9.60 | 1.77 | 37.65 | 18.11 |
| | LoRA | 71.14 | 87.90 | 81.23 | 82.27 | 91.98 | 88.00 | 93.65 | 87.63 | 85.48 |
| | LoRA+ | 73.12 | 86.18 | 80.39 | 81.23 | 92.29 | 85.80 | 94.26 | 87.13 | 85.05 |
| | DoRA | 74.60 | 89.30 | 79.90 | 80.40 | 90.50 | 85.80 | 95.50 | 85.60 | 85.20 |
| | AdaLoRA | 71.93 | 88.68 | **81.99** | 83.19 | 93.22 | **89.40** | **96.26** | 88.71 | 86.67 |
| | DiRA | **76.12** | **90.91** | 81.68 | **83.70** | **93.56** | 88.40 | 95.78 | **89.42** | **87.45** |

Table 1 reports accuracy on eight commonsense-reasoning benchmarks for several PEFT methods. Overall, DiRA achieves the highest average accuracy for both model families: for Llama2-7B DiRA attains an average of 82.75%, outperforming LoRA, LoRA+, DoRA and AdaLoRA by +1.32%, +0.66%, +3.06% and +0.15% points respectively; for Llama3-8B DiRA reaches an average of 87.45%, beating AdaLoRA , LoRA, LoRA+ and DoRA by +0.78%, +1.97%, +2.40% and +2.25% points respectively. Both Llama variants with DiRA also notably exceed the ChatGPT baseline average (77.01%) reported in the table.

A task-level inspection shows that DiRA yields consistent improvements on several benchmarks: on Llama2-7B it is best on BoolQ, PIQA, SIQA, ARC-c and OBQA; on Llama3-8B it leads on BoolQ, PIQA, ARC-c and ARC-e (with particularly large gains on BoolQ and PIQA). There are a few exceptions where alternative methods perform slightly better (e.g., AdaLoRA attains the best SIQA on Llama3-8B and the highest HellaSwag on some settings), indicating small task-dependent differences. In summary, DiRA provides the strongest and most consistent average performance across these commonsense reasoning benchmarks while exhibiting only minor per-task variability relative to other competitive PEFT methods.

## 4.3 RESULTS ON CONVERSATIONAL TASKS

Table 2: Results on the ConvAI2 dataset, where BERT F1, BERT-R, and BERT-P denote the F1, Precision, and Recall based on the BERT score, respectively.

| Model | Method | BLEU | BERT F1 | BERT-R | BERT-P | Meteor | R-L | Average |
|-------|--------|------|---------|--------|--------|--------|-----|---------|
| | P-Tuning | 0.60 | 83.29 | 83.33 | 83.28 | 15.11 | 12.36 | 46.33 |
| | LoRA | 4.29 | 85.12 | **85.34** | 84.94 | 13.95 | 13.41 | 47.84 |
| | LoRA+ | 4.30 | 85.00 | 85.21 | 84.83 | 13.57 | 13.00 | 47.65 |
| Llama2-7B | DoRA | 1.73 | 84.18 | 84.61 | 83.81 | 11.25 | 10.41 | 46.00 |
| | AdaLoRA | 3.14 | 81.62 | 82.95 | 80.35 | **19.19** | **14.37** | 46.94 |
| | DiRA | **4.44** | **85.15** | 85.32 | **85.01** | 14.12 | 13.59 | **47.94** |
| | P-Tuning | 1.50 | 81.52 | 81.07 | 82.01 | **15.49** | 13.55 | 45.86 |
| | LoRA | 4.78 | 84.69 | 84.22 | 85.21 | 13.80 | 13.75 | 47.74 |
| | LoRA+ | 4.86 | **84.75** | 84.27 | **85.27** | 14.3 | 13.79 | 47.87 |
| Llama3-8B | DoRA | 2.29 | 84.32 | 84.06 | 84.62 | 12.63 | 11.78 | 46.62 |
| | AdaLoRA | 4.80 | 84.70 | 84.35 | 85.08 | 14.44 | 13.99 | 47.89 |
| | DiRA | **4.88** | 84.70 | **85.30** | 85.10 | 14.64 | **14.02** | **48.11** |

In Table 2, DiRA delivers the best overall performance across both model sizes. For Llama2-7B, DiRA attains the highest Average (47.94%), improving over LoRA by $+0.10\%$, LoRA+ by $+0.29\%$, AdaLoRA by $+1.00\%$, P-Tuning by $+1.61\%$, and DoRA by $+1.94\%$. It also achieves the top scores on BLEU (4.44%), BERT F1 (85.15%), and BERT-P (85.01%), while LoRA holds the best BERT-R (85.34%) and AdaLoRA leads on Meteor (19.19%) and ROUGE-L (14.37%). For Llama3-8B, DiRA again yields the highest Average (48.11%), outperforming LoRA/LoRA+/AdaLoRA by $+0.37\%/+0.24\%/+0.22\%$, and reaches the best BLEU (4.88%), BERT-R (85.30%), and ROUGE-L (14.02%); LoRA+ slightly leads on BERT F1 (84.75%) and BERT-P (85.27%), and P-Tuning achieves the highest Meteor (15.49%). Overall, these results indicate that applying nuclear-norm regularization to weight deltas with dynamic rank adaptation improves conversational generation, boosting surface-overlap metrics (BLEU, ROUGE-L) while maintaining competitive semantic similarity (BERTScore).

## 5 RANK LANDSCAPE AND CATASTROPHIC FORGETTING: ANALYSIS AND A MITIGATION STRATEGY

In continual learning, catastrophic forgetting remains a central challenge. While its effects on performance are well-documented, the underlying structural changes within the model are less understood. Answering the question of *how* a model forgets is key to developing more robust continual learning systems. In this section, we leverage DiRA, with its layer-wise adaptive rank capabilities, as a powerful scientific probe to first uncover a fundamental two-phase dynamic of forgetting and recovery. We then introduce a novel mitigation strategy inspired directly by this scientific discovery, demonstrating a complete cycle from observation to intervention.

To dissect the forgetting process, we move beyond simple performance metrics to analyze the model's internal structural changes. Our experiments follow a classic three-stage continual learning procedure (McCloskey & Cohen, 1989; Kirkpatrick et al., 2017; Zenke et al., 2017), designed to induce and subsequently recover from forgetting. To instantiate this paradigm, we define two tasks from distinct domains: Task A is a commonsense reasoning task (Common170k) (Hu et al., 2023), and Task B is a mathematical reasoning task (GSM8K) (Cobbe et al., 2021). The significant domain gap between these tasks is intentionally chosen to induce a strong interference effect. The procedure unfolds as follows:

1. **Task A (Initial):** We first train the model on Task A to establish a baseline of knowledge.

2. **Task B (Interference):** We then continue training the same model on Task B. At this stage, the model's performance on Task A is expected to degrade significantly, exhibiting catastrophic forgetting.

3. **Task A (Recovered):** Finally, starting from the state where Task A is forgotten, we retrain the model on Task A to observe the recovery process.

## 5.1 Forgetting and Rank Landscape Guided Reparameterization (RLGR)

We define the model's **rank landscape** as a line plot illustrating the change in rank of the weight-update matrix across layers. Specifically, the plot's x-axis represents the layers of the model, and the y-axis represents the corresponding layer's effective rank.

The weight-update matrix, $\Delta W$, refers to the update induced by LoRA or the difference between the weights before and after fine-tuning. For each layer $l$, we compute the singular values $\sigma_i^l$ of its matrix $\Delta W^l$. The effective rank for that layer, $k_l$, is defined as the number of singular values greater than a specific threshold $\tau$:

$$k_l(\tau) = \#\{i : \sigma_i^l > \tau\}$$

In this work, we set the threshold to $\tau = 0.001$.

Table 3: Total spectral energy across all layers for different values of $\tau$.

| $\tau$ | Task A (Initial) | Task B (Interference) | Task A (Recovered) |
|---|---|---|---|
| 0.005 | 84.52 | 98.42 | 81.15 |
| 0.001 | **95.62** | **99.94** | **92.93** |

**Energy-based justification for $\tau = 0.001$.** To justify this absolute cutoff we evaluate the fraction of spectral (squared) energy retained by singular values above $\tau$:

$$E_l(\tau) \;=\; \frac{\sum_{i:\sigma_i^l > \tau}(\sigma_i^l)^2}{\sum_{i=1}^r (\sigma_i^l)^2}.$$

Table 3 reports the total spectral energy retained across all layers for candidate thresholds; choosing $\tau = 0.001$ preserves over 90% of the energy across stages, and is therefore adopted as our working cutoff.

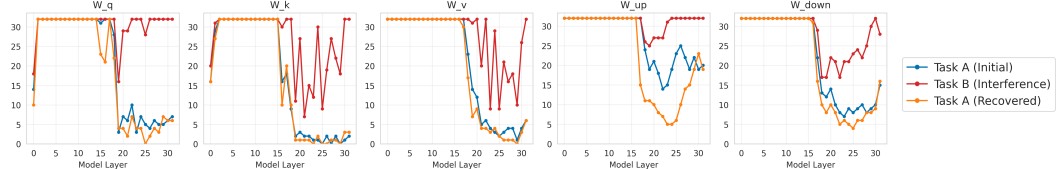

Figure 1: Rank landscape for Task A (Initial), Task B (Interference) and Task A (Recovered).

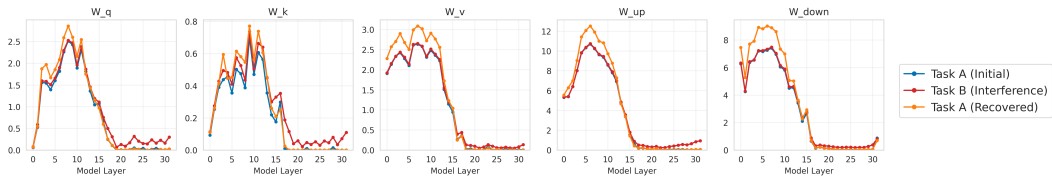

Figure 2: Nuclear norm across layers for Task A (Initial), Task B (Interference) and Task A (Recovered).

We introduce a visualization that makes forgetting explicit(Figure 1 and Figure 2): changes in the rank landscape are directly linked to performance fluctuations, enabling actionable interpretations and operational metrics for future studies of model forgetting. Specifically, we observe a pronounced increase in rank at the model's upper layers, while layer-wise nuclear norms remain nearly unchanged after interference. After recovery, the original rank landscape is largely restored; nuclear

Table 4: Forgetting on Task A and Task B. Task A reports Commonsense Reasoning test results; Task B reports GSM8K test results.

| Model | Task A | Task B |
|---|---|---|
| Task A (Initial) | **87.45** | 8.49 |
| Task B (Interference) | 12.16 | **57.14** |
| Task A (Recovered) | 87.13 | 12.96 |

norms are slightly higher than in both Initial and Interference stages (Figure 2), consistent with adding the degrees of freedom needed to rebuild the low-rank structure while absorbing limited representations from Task B.

Mechanistically, catastrophic forgetting arises when updates for a new task interfere with weights important to previous tasks; protecting those weights, as in Elastic Weight Consolidation (EWC) (Kirkpatrick et al., 2017), mitigates interference. Replay-based methods (e.g., iCaRL) (Rebuffi et al., 2017) and gradient-constraint methods (e.g., GEM, MER) (Lopez-Paz & Ranzato, 2017; Riemer et al., 2018)offer complementary remedies, but they operate at sample or gradient level rather than at the structural level.

Motivated by this evidence, we adopt a structure-first strategy: guide parameter updates along spaces aligned with Task A's low-rank prior, instead of adapting parameters freely and only later attempting to restore structure. Concretely, using the rank landscape learned at Task A (Initial), RLGR identifies layers whose effective rank is 0 or 1 ($W_q, W_k, W_v, W_{up}, W_{down}$). For these layers, we zero the corresponding LoRA_B adapters in Task B (Interference) before fine-tuning on a small subset of Task A data. This guided reparameterization reduces interference and achieves a better trade-off than naive small-data retraining—yielding improved Task A recovery with smaller Task B degradation(See Figure 3).

## 5.2 EXPERIMENTAL VALIDATION OF RLGR

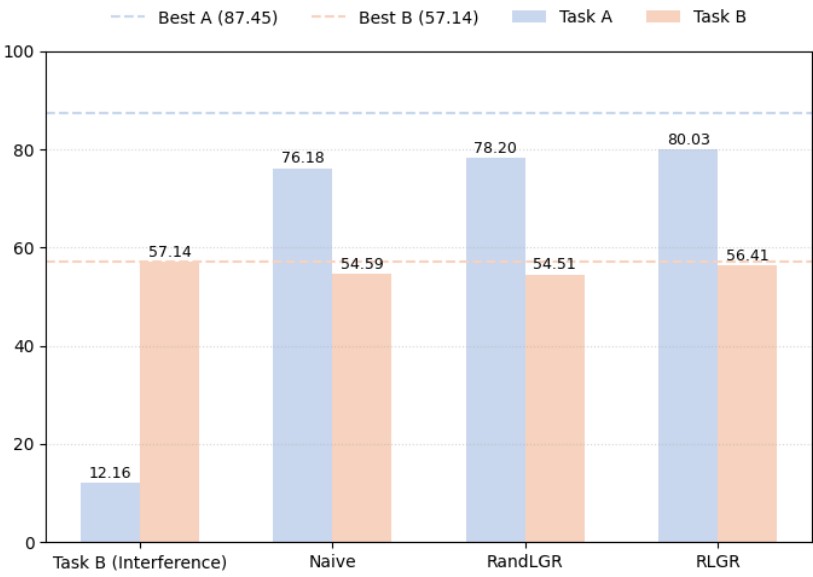

Figure 3: Accuracy comparison of recovery strategies on Task A and Task B.

In Table 4 we observe severe catastrophic forgetting on Task A. Although fine-tuning on Task B substantially improves performance on Task B, it causes a dramatic drop in Task A performance (from 87.45% to 12.16%). Retraining extensively on Task A restores Task A performance (Task A (Recovered)), but this recovery comes at the expense of Task B, which is forgotten again.

To validate the effectiveness of our proposed **RLGR** strategy, we start from the Task B (Interference) checkpoint and attempt to recover Task A performance while minimizing damage to Task B. We compare RLGR against two baselines: (i) **Naive**, which fine-tunes directly on a small subset of Task A; and (ii) **RandLGR**, which randomly selects layers to zero out and then fine-tunes on the same small subset of Task A. For these experiments we use 8 training examples from Task A, a batch size of 8, and train for 2 epochs.

As shown in Figure 3, Naive restores A to 76.18% (approximately 87.11% of the best A) and 54.59% (B). RandLGR improves slightly to 78.20% (A) and 54.51% (B). Our RLGR yields the best trade-off, recovering A to 80.03% (91.5% of the best A) while largely preserving B (56.41%, approximately 98.7% of the best B), demonstrating more effective interference recovery with minimal collateral forgetting.

## 6 CONCLUSION

We introduce DiRA, a simple end-to-end PEFT method that dynamically adapts the rank of each layer during fine-tuning. Extensive experiments demonstrate that DiRA substantially improves performance across a variety of tasks. Further study in the continual learning reveals a mechanistic link between forgetting and the model's rank landscape: forgetting is frequently accompanied by pronounced changes in the rank landscape, and exploiting this relationship can mitigate catastrophic forgetting. In summary, DiRA not only offers a practical, low-overhead means of allocating capacity to crucial locations and improving model performance, but also proposes a novel analysis and extension for using rank-based priors to reduce forgetting in continual learning.

## 7 ETHICS STATEMENT

This study does not involve human participants or animal subjects. We exclusively use publicly available datasets and strictly comply with their licenses and terms of use. No personally identifiable information (PII) is collected, processed, or disclosed. Given the nature and source of the data, institutional review board (IRB) approval was not required. We assessed potential ethical risks—including privacy, security, fairness, and misuse—and did not identify material concerns beyond those commonly associated with standard academic research. To support transparency and reproducibility, we plan to release our code under an appropriate open-source license together with documentation and a responsible-use notice; all released artifacts exclude sensitive or identifiable information. If future extensions of this work involve human data or sensitive applications, we will obtain prior ethics approval and implement appropriate safeguards.

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

## A   THE USE OF LARGE LANGUAGE MODELS(LLMS)

We used LLMs solely to assist with language polishing (grammar, clarity, and style); they made no contributions to the research design, experiments, analyses, or technical claims. All suggested edits were reviewed, edited, and approved by the authors, and any change affecting meaning or numerical content was manually verified.

## B   ABLATION

### B.1   IMPACT OF DIFFERENT CHOICES OF REGULARIZATION COEFFICIENT

Table 5: Performance comparison between different choices of regularization coefficient $\lambda$ defined in Eq. 5. Cosine indicates that the $\lambda$ follows a cosine decay schedule during training, while Linear indicates that it decays linearly. Both schedules start from $\lambda = 1 \times 10^{-4}$.

| Model | $\lambda$ | BoolQ | PIQA | SIQA | ARC-c | ARC-e | OBQA | HellaS | WinoG | Average |
|---|---|---|---|---|---|---|---|---|---|---|
| | $1 \times 10^{-3}$ | 68.96 | 88.57 | 81.47 | 82.08 | 92.59 | 86.40 | 94.46 | 87.21 | 85.22 |
| | $5 \times 10^{-4}$ | 70.18 | 89.12 | 81.58 | 83.53 | 93.43 | 87.60 | 95.26 | 88.40 | 86.14 |
| Llama3-8B | $5 \times 10^{-5}$ | 71.56 | 87.64 | 82.14 | 83.28 | 93.52 | **89.80** | 95.63 | 88.16 | 86.47 |
| | Cosine | 75.66 | 89.45 | **83.42** | **83.78** | 93.47 | 88.80 | **96.07** | 88.16 | 87.35 |
| | Linear | **76.12** | **90.91** | 81.68 | 83.70 | **93.56** | 88.40 | 95.78 | **89.42** | **87.45** |

In this section we study the impact of different choices for the regularization coefficient $\lambda$ (defined in Eq. 5) on commonsense reasoning benchmarks. Table 5 reports results for Llama3-8B under three settings: constant constant $\lambda \in \{1 \times 10^{-3}, 5 \times 10^{-4}, 5 \times 10^{-5}\}$, a cosine decay schedule (Cosine), and a linear decay schedule (Linear). All runs use the same training protocol (optimizer, learning rate, and number of epochs).

As shown in Table 5, both decay schedules outperform the constant $\lambda$ baseline: the Linear schedule achieves the best average of 87.45%, narrowly ahead of Cosine at 87.35%, and both surpass the best constant setting $5 \times 10^{-5}$ (86.47%) by +0.98% and +0.88% points, respectively, while an overly strong constant $1 \times 10^{-3}$ degrades performance to 85.22%. By task, Linear is strongest on BoolQ, PIQA, ARC-e, and WinoG; Cosine leads on SIQA, ARC-c, and HellaSwag; and OBQA peaks with the weaker constant $5 \times 10^{-5}$. These results suggest that using schedule strategy during training offers a more robust trade-off than keeping $\lambda$ fixed.

### B.2   IMPACT OF DIFFERENT CHOICES OF NUCLEAR NORM REGULARIZATION TERM

We compare two practical instantiations of nuclear-norm–inspired regularization for LoRA updates. **DiRA** applies the rank-1 decomposition penalty and augments the task loss with a per-component product regularizer:

$$\mathcal{L}_{\text{total}} = \mathcal{L}_{\text{task}} + \lambda \sum_{l=1}^{k} \sum_{i=1}^{r} \|B_{:,i}^l\|_2 \|A_{i,:}^l\|_2,$$

which follows the rank-1 decomposition representation of the nuclear norm (Eq. 3). In contrast, **DiRA_var** uses the factorized variational proxy

$$\mathcal{L}_{\text{total}} = \mathcal{L}_{\text{task}} + \frac{\lambda}{2} \sum_{l=1}^{k} \left( \|B^l\|_F^2 + \|A^l\|_F^2 \right),$$

corresponding to Eq. 2. Both variants are trained with the same maximum rank $r$ and compute budget.

Table 6 reports downstream performance: on Llama2-7B, DiRA attains an average of 82.75% versus 82.20% for DiRA_var (+0.55%), with DiRA outperforming on BoolQ, PIQA, SIQA (+2.15%), ARC-c and OBQA, while DiRA_var is marginally better on ARC-e, HellaSwag and WinoGrad; on Llama3-8B the two methods are very close (averages 87.45% vs. 87.31%, $\Delta = +0.14\%$), where DiRA

Table 6: Performance comparison between different choices of nuclear norm regularization term.

| Model | Method | BoolQ | PIQA | SIQA | ARC-c | ARC-e | OBQA | HellaS | WinoG | Average |
|-------|--------|-------|------|------|-------|-------|------|--------|-------|---------|
| Llama2-7B | DiRA$_{var}$ | 72.39 | 84.93 | 78.30 | 74.40 | **88.17** | 84.80 | **88.71** | **85.87** | 82.20 |
| | DiRA | **72.84** | **85.47** | **80.45** | **75.68** | 88.13 | **85.80** | 87.97 | 85.64 | **82.75** |
| Llama3-8B | DiRA$_{var}$ | 75.84 | 89.66 | **82.65** | 83.28 | 93.48 | **88.60** | 95.69 | 89.27 | 87.31 |
| | DiRA | **76.12** | **90.91** | 81.68 | **83.70** | **93.56** | 88.40 | **95.78** | **89.42** | **87.45** |

leads on BoolQ, PIQA (+1.25%), ARC-c and ARC-e, and DiRA$_{var}$ is slightly stronger on SIQA (+0.97%) and OBQA (+0.20%).

Overall, when tuned to the same parameter budget the two penalties yield comparable end-task accuracy, with DiRA showing a small but consistent advantage in average score and on several individual tasks; these results suggest that the low-rank inductive bias induced by nuclear norm regularization—rather than the precise algebraic form of the proxy—is the primary driver of the observed gains. The specific penalty, however, influences secondary properties such as the sparsity/concentration of learned rank-1 components, optimization stability and per-task variability.

### B.3 SINGULAR VALUE ANALYSIS OF DiRA AND LoRA

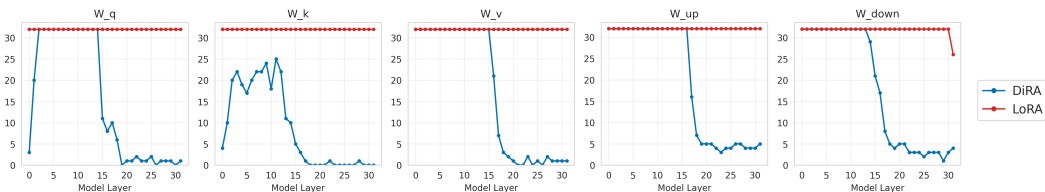

Figure 4: Rank landscape for DiRA and LoRA.

Figure 4 shows the number of singular values greater than 0.001 for representative weight matrices $(W_q, W_k, W_v, W_{up}, W_{down})$ across layers after fine-tuning Llama3-8B on the commonsense reasoning datasets. Unlike fixed-rank LoRA (red), which remains near the maximum rank for almost all layers, DiRA (blue) exhibits strong layer-wise variation: a small number of layers retain many non-negligible singular values while most layers have an effective rank near zero. This rank landscape indicates that DiRA dynamically allocates effective rank according to each layer's utility, rather than applying a uniform low-rank update everywhere.

Table 7: Performance comparison between DiRA$_{front}$, DiRA$_{back}$ and DiRA (original) across commonsense reasoning datasets.

| Method | BoolQ | PIQA | SIQA | ARC-c | ARC-e | OBQA | HellaS | WinoG | Average |
|--------|-------|------|------|-------|-------|------|--------|-------|---------|
| DiRA$_{front}$ | 56.51 | 68.34 | 30.04 | 15.10 | 13.72 | 3.40 | 11.88 | 11.92 | 26.36 |
| DiRA$_{back}$ | **76.39** | 22.09 | 78.40 | 82.94 | 93.18 | 87.40 | 38.76 | 81.13 | 70.04 |
| DiRA | 76.12 | **90.91** | **81.68** | **83.70** | **93.56** | **88.40** | **95.78** | **89.42** | **87.45** |

To verify that the learned per-layer ranks are functionally meaningful, we perform a targeted ablation. Let DiRA denote the original trained model; define DiRA$_{front}$ by zeroing the DiRA updates for layers 1–16 (i.e., setting the learned updates of $W_q, W_k, W_v, W_{up}, W_{down}$ to zero in those layers while leaving layers 17–32 unchanged), and define DiRA$_{back}$ by zeroing the DiRA updates for layers 17–32 (layers 1–16 unchanged). Results are reported in Table 7. The original DiRA model achieves an average score of 87.45%. Removing the learned updates in the upper half of the network (DiRA$_{back}$) reduces the average to 70.04%, whereas removing the learned updates in the lower half (DiRA$_{front}$) causes a severe collapse to 26.36%. The asymmetric damage produced by these ablations matches the rank landscape in Figure 4 and demonstrates that DiRA concentrates its limited rank capacity in the layers that are most critical for downstream performance. In short, the

learned per-layer ranks are not an incidental numerical artifact but encode functionally important, layer-adaptive information.

### B.4 SINGULAR VALUE ANALYSIS OF DIFFERENT DATASETS

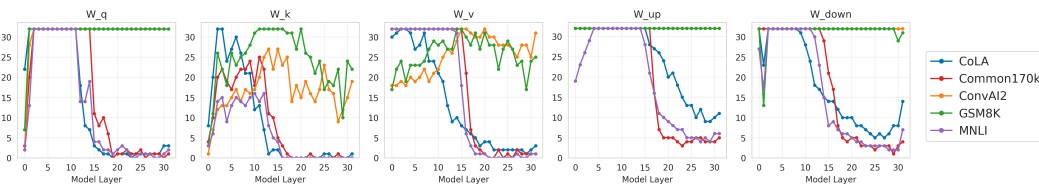

Figure 5: Rank landscape for different datasets.

We apply DiRA to learn layer-wise low-rank updates on five datasets and analyze the resulting rank allocation by counting singular values of each layer's adaptation matrix that exceed 0.001 (Figure 5). The plotted curves use CoLA (blue) (Warstadt et al., 2019), Common170k (red) (Hu et al., 2023), ConvAI2 (orange) (Dinan et al., 2019), GSM8K (green) (Cobbe et al., 2021) and MNLI (purple) (Williams et al., 2017). CoLA, Common170k and MNLI show high rank in the lower layers and sharply reduced rank in higher layers, whereas GSM8K and ConvAI2 maintain relatively high rank across both lower and higher layers.

This pattern aligns with prior findings (Tenney et al., 2019; Jawahar et al., 2019; Hewitt & Manning, 2019; Rogers et al., 2021) that syntactic or surface-level information is primarily stored in lower transformer layers while semantic and reasoning knowledge is concentrated higher up: CoLA/Common170k/MNLI emphasize low-layer adaptation, while GSM8K/ConvAI2 require adaptation throughout the network. The result demonstrates that DiRA discovers task-dependent, non-uniform rank landscape and supports the practical recommendation that PEFT methods should allow layer-wise, dataset-sensitive allocation of adaptation capacity.

## C TRAINING DETAILS

### C.1 TRAINING HYPERPARAMETERS

Table 8: Hyperparameters for DiRA.

| Hyperparameter | Value |
|---|---|
| Optimizer | AdamW |
| Weight Decay | 0 |
| Base Model | [Llama2-7B, Llama3-8B] |
| Learning Rate | 0.0001 |
| Rank r | 32 |
| Regularization Coefficient $\lambda$ | 0.0001 |
| Decay Schedule of $\lambda$ | Linear |
| Batch Size | 32 |
| Warm Up | 100 steps |
| Target Modules | q proj, k proj, v proj, up proj, down proj |
| Evaluation Steps | Every 80 steps |

Table 8 lists the hyperparameters used to tune Llama2-7B and Llama3-8B with DiRA on two tasks: commonsense reasoning and ConvAI2. Both tasks share the same hyperparameter configuration except for the number of epochs (three for commonsense reasoning, one for ConvAI2). Each experiment was run separately, with a single run per model for each tested value of $r$; the best result was selected according to validation loss. Baseline methods reused these configurations where applicable. Note that different PEFT methods may require distinct hyperparameter choices to attain comparable numbers of trainable parameters.

## C.2 Details of Commonsense Reasoning Dataset

Table 9: The details of commonsense reasoning datasets.

| Dataset | Data Number | Type |
|---|---|---|
| Train | 170,300 | Mixed |
| Validation | 120 | Mixed |
| Test | | |
| BoolQ | 3270 | Yes/No |
| PIQA | 1830 | Option |
| SIQA | 1954 | Option |
| HellaSwag | 10042 | Option |
| WinoGrande | 1267 | Option |
| ARC-e | 2376 | Option |
| ARC-c | 1172 | Option |
| OBQA | 500 | Option |

As shown in Table 10, the training set comprises eight sub-tasks totaling 170,300 examples, the validation set contains a random sample of 120 examples, and the test set covers the same eight sub-tasks; evaluation is performed using a single trained model.

## C.3 Details of the ConvAI2 Dataset

Table 10: The details of the ConvAI2 dataset.

| Data Split | Utterances | Dialogues | Personas |
|---|---|---|---|
| Train | 131,438 | 17,878 | 1155 |
| Test | 7,801 | 1,000 | 100 |

We conduct experiments on the ConvAI2 dataset (Dinan et al., 2019), a standard benchmark for open-domain dialogue generation. As reported in Table 8, ConvAI2 contains 17,878 training and 1,000 testing multi-turn conversations collected from crowdworkers. Each dialogue includes persona profiles—four to five sentences describing each speaker's background—together with the conversational history between the two interlocutors. Following (Liu et al., 2020; Song et al., 2021), we adopt the self-persona setting: only the speaking interlocutor's persona is revealed, while the partner's persona is withheld.

## D Evaluation on Transfer Learning Tasks

Figures 6 summarizes cross-task transfer when the model is fine-tuned on one dataset (rows) and evaluated on eight tasks (columns). We report four key observations.

**Overall effect.** DiRA substantially strengthens cross-task transfer while preserving in-task performance. On the diagonal (in-distribution evaluation), DiRA is on par with or slightly better than LoRA: e.g., ARC-e 92.8% vs. 92.2%, SIQA 82.7% vs. 82.6%, and Winogrande 87.8% vs. 87.7%, with only a minor drop on OBQA (86.2% vs. 87.8%). Off-diagonal (out-of-distribution) gains are pronounced: BoolQ $\rightarrow$ ARC-e improves from 24.8% to 47.1%, BoolQ $\rightarrow$ OBQA from 8.8% to 26.8%; Winogrande $\rightarrow$ PIQA from near-zero to 68.2%; HellaSwag $\rightarrow$ BoolQ from 0 to 62.2%, and HellaSwag $\rightarrow$ Winogrande from 0 to 52.2%. These results indicate that DiRA enhances adapter reuse across tasks without sacrificing source-task accuracy.

**Task clusters and asymmetry.** We observe a "reasoning cluster" comprising SIQA, ARC-c, ARC-e, and OBQA, where mutual transfer is consistently strong under both methods, and further strengthened by DiRA (e.g., SIQA $\rightarrow$ ARC-e: 88.6% $\rightarrow$ 90.2%). In contrast, PIQA, HellaSwag, and

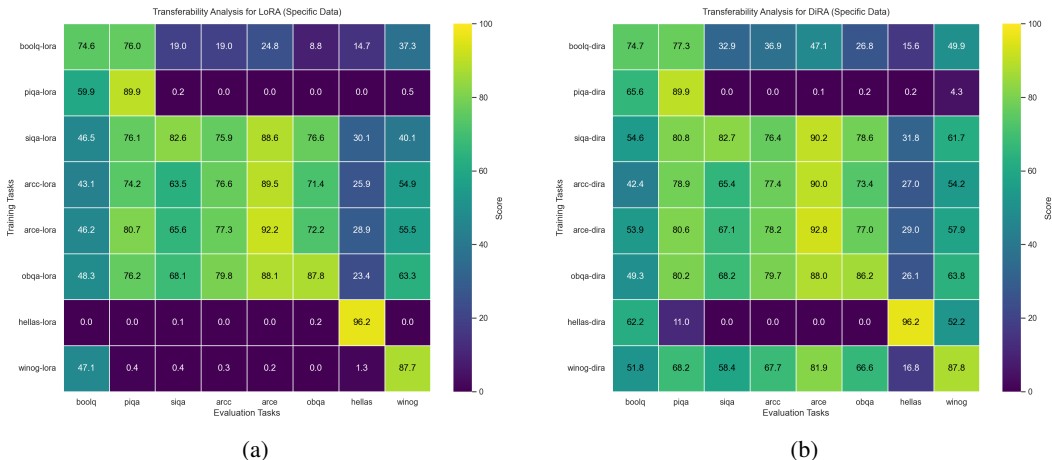

Figure 6: Transferability analysis for (a) LoRA and (b) DiRA. Each row represents the performance of a fine-tuned model trained on a specific task dataset when evaluated across eight different tasks.

Winogrande exhibit specialization under LoRA (limited transfer to other tasks). DiRA mitigates the isolation for HellaSwag and Winogrande but PIQA remains relatively asymmetric: it transfers weakly to most other tasks despite modest improvements (e.g., to BoolQ from 59.9% to 65.6%).

**Methodological comparison.**   While LoRA achieves similar ceilings on the source tasks, it generally underperforms off-diagonal, with some transfers nearly failing (notably from Winogrande and HellaSwag). DiRA consistently yields positive transfer across many mismatched pairs and does so without notable regressions on the training task.

**Practical implications.**   (i) When downstream tasks are uncertain or broad generalization is desired, prefer DiRA: it offers significant cross-task gains with no meaningful loss on the source task. (ii) If the goal is to maximize a single, known distribution (e.g., HellaSwag or PIQA), both methods perform similarly on the diagonal; LoRA remains a lightweight option. (iii) For residual island behaviors (especially PIQA), multi-source training, contrastive/relational regularization, or task-mixing curricula may further encourage shared representations and reduce negative transfer.

In summary, under our setting, DiRA emerges as a more generally transferable and robust parameter-efficient fine-tuning strategy: it maintains source-task accuracy while markedly improving cross-task generalization and reusability.

# E   THEORETICAL DERIVATION OF AN UPPER BOUND ON THE NUCLEAR NORM

**Rank-1 norm identity.**   Let $u \in \mathbb{R}^{d_2}$ and $v \in \mathbb{R}^{d_1}$. For the rank-1 matrix $uv^\top$, all three matrix norms—nuclear, Frobenius, and spectral—coincide and equal the product of the vector Euclidean norms:

$$\|uv^\top\|_* = \|uv^\top\|_F = \|uv^\top\|_2 = \|u\|_2 \|v\|_2.$$

This follows because $uv^\top$ has exactly one nonzero singular value, which is $\sigma_1 = \|u\|_2 \|v\|_2$.

**Upper bound via triangle inequality.**   Writing the LoRA update as a sum of rank-1 components,

$$\Delta W^l = \sum_{i=1}^{r} B_{:,i}^l A_{i,:}^l,$$

the triangle inequality of the nuclear norm yields

$$\left\|\Delta W^l\right\|_* = \left\|\sum_{i=1}^{r} B_{:,i}^l A_{i,:}^l\right\|_* \leq \sum_{i=1}^{r} \left\|B_{:,i}^l A_{i,:}^l\right\|_*.$$

Applying the rank-1 identity to each term gives a computable upper bound:

$$\left\|\Delta W^l\right\|_* \leq \sum_{i=1}^{r} \|B^l_{:,i}\|_2 \|A^l_{i,:}\|_2.$$

**Practical training objective.**   Replacing the (costly) nuclear norm by its (cheap) upper bound, we arrive at the following surrogate objective:

$$\mathcal{L}_{\text{total}} = \mathcal{L}_{\text{task}} + \lambda \sum_{l=1}^{k} \left\|\Delta W^l\right\|_* = \mathcal{L}_{\text{task}} + \lambda \sum_{l=1}^{k} \sum_{i=1}^{r} \|B^l_{:,i}\|_2 \|A^l_{i,:}\|_2. \tag{6}$$

This surrogate preserves the low-rank inductive bias of nuclear norm regularization while avoiding large SVDs: each penalty term reduces to a product of two vector $\ell_2$ norms. In practice, this encourages many rank-1 components to shrink toward zero, driving the learned update $\Delta W^l$ to a data-adaptive (dynamic) rank without fixing $r$ a priori.

**Notation.**   $\|\cdot\|_*$ denotes the nuclear norm, $\|\cdot\|_F$ the Frobenius norm, and $\|\cdot\|_2$ the Euclidean norm for vectors (and the spectral norm for matrices when explicitly stated). In Eq. 6, $\|\cdot\|_2$ is applied to vectors $B^l_{:,i}$ and $A^l_{i,:}$.

# F   ANALYSIS ON DIFFERENT MODELS

We conduct a comprehensive analysis of the Qwen2.5 models at various scales to investigate the relationship between performance and internal structure. As shown in Table 11, we evaluate four models from the Qwen2.5 series—1.5B, 7B, 14B, and 32B—on a suite of commonsense reasoning datasets. The results unequivocally demonstrate that model performance scales directly with size. As the parameter count increases from the Qwen2.5-1.5B to the Qwen2.5-32B model, the average score across all benchmarks rises significantly from 81.34 to 92.86. This scaling trend confirms that larger models possess substantially stronger capabilities for commonsense reasoning tasks.

Table 11: Performance comparison between Qwen2.5-1.5B, Qwen2.5-7B, Qwen2.5-14B and Qwen2.5-32B across commonsense reasoning datasets.

| Model | BoolQ | PIQA | SIQA | ARC-c | ARC-e | OBQA | HellaS | WinoG | Average |
|---|---|---|---|---|---|---|---|---|---|
| Qwen2.5-1.5B | 66.39 | 83.46 | 76.46 | 79.01 | 91.29 | 84.60 | 90.36 | 79.16 | 81.34 |
| Qwen2.5-7B | 64.70 | 90.53 | 81.22 | 90.02 | 96.17 | 93.20 | 95.80 | 89.58 | 87.65 |
| Qwen2.5-14B | **78.38** | 94.29 | 85.01 | 94.19 | 98.02 | 96.60 | 97.40 | 92.66 | 92.07 |
| Qwen2.5-32B | 78.23 | **94.78** | **85.67** | **95.90** | **98.70** | **97.40** | **97.60** | **94.63** | **92.86** |

To further understand the underlying factors driving this performance improvement, we analyze the singular value landscape of key weight matrices within the models, as depicted in Figure 7. A clear trend is the increase in network depth with model scale: the 1.5B/7B models have 28 layers, the 14B model has 48 layers, and the 32B model extends to 64 layers. Furthermore, while smaller models exhibit a relatively flat effective rank landscape across layers, suggesting more homogeneous layer functions, larger models like the 14B and 32B versions show more pronounced fluctuations. This volatility suggests a higher degree of functional specialization has emerged among the layers in larger-scale models.

A particularly noteworthy observation is the non-linear relationship between effective rank and model scale. Despite being the top-performing model, the Qwen2.5-32B model consistently exhibits a lower effective rank across most of its layers compared to the 14B model. This suggests that as models grow to a very large scale, they may adopt more efficient and sparse parameterization strategies to encode knowledge, rather than simply increasing the intrinsic rank of each layer. This finding has significant practical implications for model fine-tuning. For instance, when applying low-rank adaptation techniques such as LoRA to very large-scale models, it may be possible to configure the adapters with a much smaller rank than typically assumed, thereby achieving substantial computational savings while preserving model performance. This insight into the parameterization efficiency of large models opens up new avenues for future model compression and efficient fine-tuning methodologies.

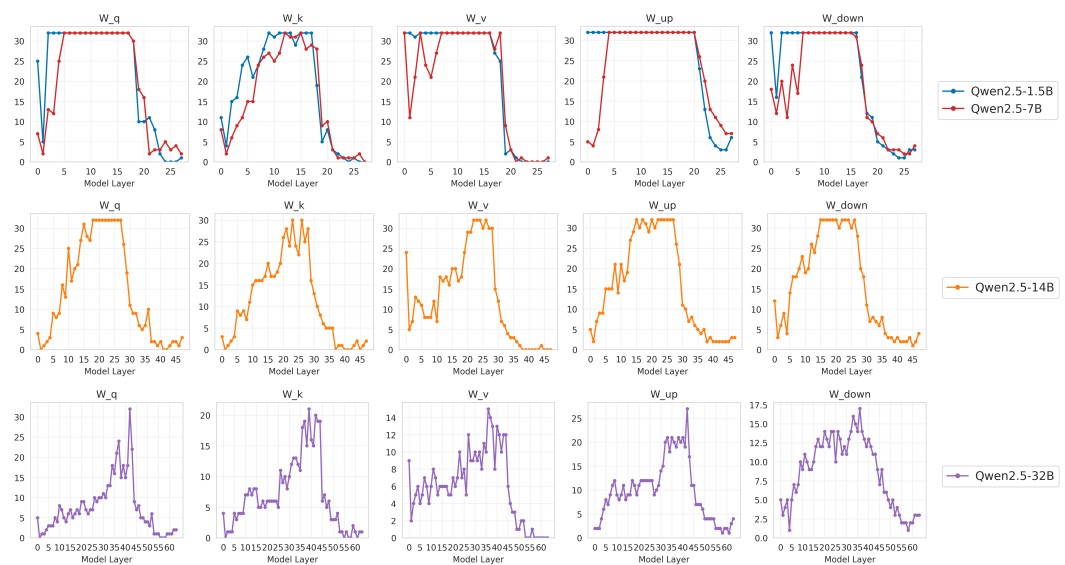

Figure 7: Rank landscape for Qwen2.5-1.5B, Qwen2.5-7B, Qwen2.5-14B and Qwen2.5-32B.

## G    ANALYSIS ON DIFFERENT RANK BUDGETS

Table 12: Performance comparison between different rank $r$ on Llama3-8B across commonsense reasoning datasets.

| Rank | BoolQ | PIQA | SIQA | ARC-c | ARC-e | OBQA | HellaS | WinoG | Average |
|------|-------|------|------|-------|-------|------|--------|-------|---------|
| 32   | 76.12 | **90.91** | 81.68 | 83.70 | 93.56 | 88.40 | 95.78 | 89.42 | 87.45 |
| 64   | 75.90 | 90.64 | **82.91** | **85.75** | **94.02** | 87.80 | 96.39 | 89.82 | 87.90 |
| 128  | **77.09** | 90.64 | 82.40 | 83.87 | 93.81 | **90.20** | **96.41** | **89.98** | **88.05** |

In our preceding analysis (see Figure 4), we established the core distinction of our proposed method, DiRA, from fixed-rank approaches like LoRA: its ability to dynamically and non-uniformly allocate rank across different model layers. This section builds upon that finding, presenting a deeper investigation into how DiRA's performance and internal rank allocation strategy adapt when provided with different total rank budgets, denoted as $r$.

First, we evaluated the performance of Llama3-8B when configured with total rank budgets of $r = 32, 64$, and $128$. As detailed in Table 12, the results demonstrate a clear and positive correlation between the rank budget and model performance. As $r$ increases from 32 to 128, the average score on commonsense reasoning tasks steadily improves from 87.45 to 88.05. This outcome aligns with our expectations, as a larger parameter budget provides greater expressive capacity for the model during fine-tuning.

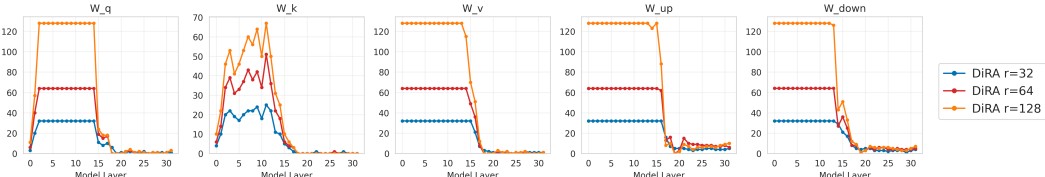

Figure 8: Rank landscape for different rank.

Figure 8 provides a visualization of how DiRA intelligently distributes these varying budgets. A key finding is that regardless of the total budget, DiRA learns a remarkably consistent allocation pattern.

It consistently concentrates the vast majority of the rank within the first approximately 15 layers of the network, particularly for the $W_q$, $W_v$, and $W_{up}$ matrices, while assigning near-zero rank to the latter half of the network. This strongly indicates that DiRA can robustly identify which layers are most critical for downstream task adaptation. When the total budget is increased, DiRA does not alter this overarching strategy; instead, it proportionally scales the ranks allocated to these crucial layers. For instance, in the early layers of $W_q$, the allocated rank is observed to be approximately equal to the total budget $r$ of the given experiment. This predictable, proportional scaling behavior further validates the robustness and effectiveness of our method's design.

## H STRUCTURAL INERTNESS OF LORA UNDER CONTINUAL LEARNING

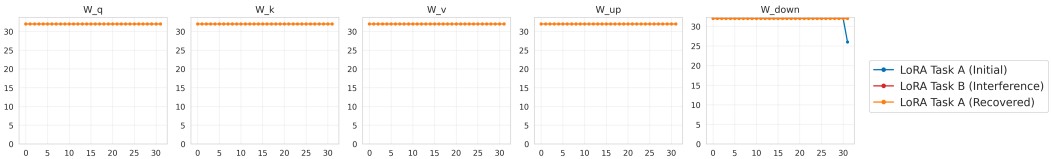

Figure 9: Rank landscape for LoRA Task A (Initial), LoRA Task B (Interference) and LoRA Task A (Recovered).

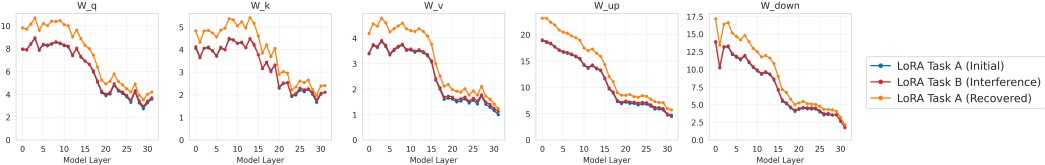

Figure 10: Nuclear norm across layers for LoRA Task A (Initial), LoRA Task B (Interference) and LoRA Task A (Recovered).

We replaced the DiRA adapter with a standard LoRA adapter and measured the same structural properties: the count of significant singular values (effective rank) and the nuclear norm across all layers. The results are presented in Figure 9 and Figure 10.

**Analysis of Rank Landscape:** As shown in Figure 9, the rank landscape of the LoRA model remains almost perfectly static across all three stages: Initial, Interference, and Recovered. The number of singular values is clamped by LoRA's predefined rank hyperparameter and does not change in response to the task shift. This directly contrasts with the dynamic disruption and reconstruction observed with DiRA (cf. Figure 1). This structural inertness makes LoRA fundamentally blind to the underlying landscape transformations that characterize catastrophic forgetting.

**Analysis of Nuclear Norm:** As shown in Figure 10, the LoRA model exhibits clear norm conservation during the Initial (Task A) and Interference (Task B) stages, the two curves are nearly overlapped, indicating that the nuclear norm remains essentially unchanged during interference. In contrast, when returning to Task A (Recovered), the nuclear norms increase noticeably over most layers—particularly for the up/down adapters and in lower-to-middle depths. This stage-wise pattern mirrors the signature observed with DiRA in Figure 2, but the magnitude of the post-recovery inflation under LoRA is consistently larger.

These results confirm that fixed-rank methods, by their very design, cannot perceive or adapt to the profound structural dynamics of continual learning. They enforce a static view on a dynamic problem, which motivates the need for adaptive-rank methods like DiRA to both uncover these phenomena and, as we propose, to intelligently mitigate them.

