# OpenReview forum: "DiRA: Nuclear Norm Dynamic Rank Adaptation for Large Language Models"
_ICLR.cc/2026/Conference — ICLR 2026 Conference Withdrawn Submission_

### Official Review · Reviewer_UeAR · 2025-10-27

**Soundness:** 2
**Presentation:** 1
**Contribution:** 1
**Rating:** 2
**Confidence:** 4

**Summary:**

DiRA is a novel Parameter-Efficient Fine-Tuning (PEFT) framework that addresses the fixed-rank limitation of Low-Rank Adaptation (LoRA) by introducing nuclear norm-based regularization. This allows dynamic, layer-wise rank adjustment. Experimental results show that DiRA outperforms existing methods such as LoRA and AdaLoRA in commonsense reasoning and dialogue generation tasks on LLaMA-based models. Furthermore, the authors demonstrate that catastrophic forgetting in continual learning is closely linked to shifts in the rank landscape and propose RLGR as a mitigation strategy.

**Strengths:**

- Demonstrated effectiveness on LLaMA-based models.

- Easy to read and follow.

- Convincingly highlights the inefficiency of fixed-rank LoRA through theoretical and empirical arguments.

- Provides an interesting analysis of the structural link between rank variation and catastrophic forgetting in continual learning.

**Weaknesses:**

- The writing lacks clarity. Without sufficient background on continual learning (CL) and without adhering to conventional experimental setups, it is difficult to assess the actual usefulness of the proposed method. The Preliminaries section should include background on CL, and the method should be compared against LoRA-based CL approaches or relevant CL baselines. For instance, replaying small amounts of previous task data is a well-established strategy called 'rehearsal' known to mitigate catastrophic forgetting. It remains unclear whether the improvements stem from rank landscape-aware mechanisms or simply from the rehearsal effect.

- Although the paper proposes a dynamic rank allocation method, it lacks to compare against latest existing approaches such as SaLoRA [1] and DyLoRA [2]. Including these as baselines is important to fairly evaluate the contribution and effectiveness of the proposed method.

- According to [3], standard weight decay in LoRA already plays a role similar to nuclear norm regularization. This paper adopts the HiRA framework and sets weight decay to zero, which may lead to substantial differences. This raises the concern that standard LoRA may already achieve a similar effect to DiRA, and clearer justification and discussion are needed.

- While the use of nuclear norm can be interpreted from a rank allocation perspective, the method itself is relatively simple, and there is a lack of empirical analysis showing how LoRA adapts its rank during training. A deeper quantitative analysis of the adapter's rank distribution or its variation over time would strengthen the validity of the approach. Furthermore, since the adapter's rank is still capped at a fixed maximum \(r\), the method does not enable higher-rank modeling or improved computational efficiency, which limits its contribution.

- The paper does not include results on widely used NLU tasks, and comparisons with more recent LoRA variants are lacking.

- There is no analytical or empirical analysis of time/space complexity.

> [1] Li, Mingjie, et al. "SaLoRA: Safety-Alignment Preserved Low-Rank Adaptation." The Thirteenth International Conference on Learning Representations.
>
> [2] Valipour, Mojtaba, et al. "DyLoRA: Parameter-Efficient Tuning of Pre-trained Models using Dynamic Search-Free Low-Rank Adaptation." Proceedings of the 17th Conference of the European Chapter of the Association for Computational Linguistics. 2023.
>
> [3] Jang, Uijeong, Jason D. Lee, and Ernest K. Ryu. "LoRA training in the NTK regime has no spurious local minima." Proceedings of the 41st International Conference on Machine Learning. 2024.

**Questions:**

- Please refer to the Weaknesses

---

### Official Review · Reviewer_3DPA · 2025-10-29

**Soundness:** 2
**Presentation:** 2
**Contribution:** 2
**Rating:** 4
**Confidence:** 3

**Summary:**

The authors present **Dynamic Rank Adaptation (DiRA)**, a method for continual learning. They propose learning layer-adaptive ranks by penalizing the nuclear norm of the weight update matrix. Specifically, they decompose the LoRA update as a sum of rank-1 components and penalize the Frobenius norm of each per-component product. This drives entire rank-1 components to zero, allowing the effective rank of each layer to adapt organically. The authors further suggest that forgetting is connected to large changes in the model’s rank landscape and propose mitigating this via a rank-landscape prior using data from previous tasks.

DiRA is evaluated on commonsense reasoning tasks and dialogue generation across LLaMA-2-7B and LLaMA-3-8B. They also probe forgetting in commonsense reasoning and math fine-tuning.

**Strengths:**

The paper introduces a novel and well-motivated approach to adaptive rank modulation.

---

- The text is clear, and the method is introduced in a straightforward and understandable manner (first part).
- The choice of the threshold is well grounded and appropriately justified.

**Weaknesses:**

### **1. Scope and Problem Framing**

- The paper appears to address two different settings: learning capability and continual learning. Presenting both in the same work is somewhat confusing.
- The recovery stage in the continual learning setup is not well motivated.
- The experiments in the continual learning setup involve effectively one task, which does not qualify as continual learning.
- The work appears to be the beginning of an interesting direction, but not fully developed or polished. Overall, the submission feels unfinished.

---

### **2. Related Work Coverage**

- The related work section reads more like a high-level introduction; several mentioned methods are not directly relevant, and connections to this work are not clearly articulated.
- Some important related methods (e.g., MiLoRA [1] and PiSSA [2]) appear to be missing.

---

### **3. Methodological Clarity**

- RLGR is never introduced before being used. Later, the text does not clearly state what is being done algorithmically. The description “zero the corresponding LoRA B adapters in Task B (Interference) before fine-tuning on a small subset of Task A data” is not sufficiently detailed to understand.

---

### **4. Presentation and Organization**

- Portions of the main text are overly descriptive (e.g., experimental setup) and could be moved to the appendix.
- Figures 1 and 2 are insufficiently analyzed; the implications of what they show are unclear.
- Figure 3 appears pixelated, and text quality drops noticeably.
- Captions are incomplete and do not describe what the reader should observe.
- Some figures lack y-axis labels, and axis labels are generally too small to read comfortably.
- The text discusses performance fluctuations, but the plots only show rank fluctuations.

---

### **5. Experimental Evaluation**

- Computational cost analysis is missing, especially given the SVD operations involved.


[1] MiLoRA: Harnessing Minor Singular Components for Parameter-Efficient LLM Finetuning. (2025) Hanqing Wang and Yixia Li and Shuo Wang and Guanhua Chen and Yun Chen

[2] PiSSA: Principal Singular Values and Singular Vectors Adaptation of Large Language Models. (2025) Fanxu Meng and Zhaohui Wang and Muhan Zhang

**Questions:**

1. Are all experiments conducted with only one seed?
2. The proposed three-stage procedure is not standard. The “recovery” stage is typically not available in realistic continual learning scenarios. How practical is this component?
3. For Figures 1 and 2, which model are you reporting results on?
4. Why is the nuclear norm higher for lower layers?
5. Could the authors showcase connections to [1]?

[1] LoRA vs Full Fine-tuning: An Illusion of Equivalence. (2025) Reece Shuttleworth and Jacob Andreas and Antonio Torralba and Pratyusha Sharma

---

### Official Review · Reviewer_tVTm · 2025-10-31

**Soundness:** 2
**Presentation:** 3
**Contribution:** 2
**Rating:** 4
**Confidence:** 4

**Summary:**

The paper presents DiRA (Dynamic Rank Adaptation), a PEFT method that addresses the limitation of LoRA, which fixes the same rank across all layers. DiRA formulates rank allocation as an optimization problem by introducing a nuclear norm regularization term in the loss. To remain efficient, it penalizes a tractable upper bound of the nuclear norm, enabling the model to learn layer-specific effective ranks by shrinking unimportant rank-1 components during training.
Experiments on commonsense reasoning and dialogue tasks show that DiRA matches or slightly outperforms LoRA and dynamic variants such as AdaLoRA. Using DiRA as a probe for continual learning, the study finds that catastrophic forgetting corresponds to shifts in the model’s rank landscape, and introduces RLGR, a recovery method that leverages prior rank landscapes to improve knowledge retention.

**Strengths:**

The use of nuclear norm regularization to induce a dynamic, layer-wise rank is well-motivated, offering a soft alternative to the hard pruning or SVD-based importance scoring employed in methods such as AdaLoRA.

**Weaknesses:**

1. The main limitation of DiRA lies in its modest empirical improvements. Although presented as a superior PEFT method, its performance gains over AdaLoRA, a strong dynamic-rank baseline, are minimal and inconsistent across tasks.

2. The proposed RLGR strategy depends on access to “a small subset of data from previous tasks” (eight examples), effectively functioning as a data replay approach. While framed as a novel insight derived from the rank landscape analysis, this reliance on replay makes its originality and practical benefit difficult to evaluate relative to established replay- or regularization-based CL methods. Moreover, RLGR is applied post hoc—to recover performance on a forgotten task—rather than preventively mitigating forgetting during subsequent training, which limits its generality.

3. The continual learning study appears tacked on and insufficiently validated. RLGR is tested on only one task pair (Common170k → GSM8K) and compared solely against Naive and RandLGR, omitting stronger CL baselines such as EWC, LwF, or other replay methods. As a result, the CL component feels preliminary and distracts from the paper’s core contribution—an already marginally superior PEFT method.

**Questions:**

Refer to Weaknesses for related questions.

---

### Official Review · Reviewer_tpMu · 2025-11-01

**Soundness:** 2
**Presentation:** 3
**Contribution:** 2
**Rating:** 4
**Confidence:** 4

**Summary:**

This paper introduces DiRA, a novel PEFT method that enables dynamic, layer-wise rank allocation for LoRA by penalizing the nuclear norm of the weight update matrix. Beyond its performance as an efficient PEFT method, DiRA is utilized as a scientific probe to uncover a mechanism of catastrophic forgetting in continual learning.

**Strengths:**

- The nuclear-norm-style factor penalty is well motivated and avoids full SVDs.
- DiRA performs as well or better than LoRA over multiple datasets (commonsense + ConvAI2).
- Using DiRA to study catastrophic forgetting and rank dynamics is potentially impactful.

**Weaknesses:**

- Comparisons to recent rank-adaptive LoRA variants (e.g., AdaLoRA, RankAdaptor, ARD-LoRA) would be needed.
- Rank adapts per layer, but overhead is not clearly quantified vs. LoRA or SVD-based methods.
- The CL part is more exploratory than conclusive; missing LoRA variants for continual learning baselines (e.g., LoRI).
- The choice/sensitivity of the regularization strength is not well discussed.

**Questions:**

- What is the actual fine-tuning cost relative to standard LoRA? Does the per-layer dynamic structure add parameter overhead?
- For CL, is performance robust across different sequences of tasks or domains?

---

### Note · Authors · 2025-11-12

I have read and agree with the venue's withdrawal policy on behalf of myself and my co-authors.